# Cardiorespiratory Fitness as Mediator of the Relationship of Recreational Screen Time on Mediterranean Diet Score in Schoolchildren

**DOI:** 10.3390/ijerph18094490

**Published:** 2021-04-23

**Authors:** José Francisco López-Gil, Antonio García-Hermoso, Javier Brazo-Sayavera, Pedro Juan Tárraga López, Juan Luis Yuste Lucas

**Affiliations:** 1Departamento de Actividad Física y Deporte, Facultad de Ciencias del Deporte, Universidad de Murcia (UM), 30720 San Javier, Spain; 2Navarrabiomed, Complejo Hospitalario de Navarra (CHN), Universidad Pública de Navarra (UPNA), IdiSNA, 31008 Pamplona, Spain; antonio.garcia.h@usach.cl; 3Escuela de Ciencias de la Actividad Física, el Deporte y la Salud, Universidad de Santiago de Chile (USACH), Santiago 71783-5, Chile; 4Department of Sports and Computer Science, Universidad Pablo de Olavide (UPO), 41013 Seville, Spain; jbsayavera@upo.es; 5PDU EFISAL, Centro Universitario Regional Noreste, Universidad de la República (UDELAR), 40000 Rivera, Uruguay; 6Departamento de Ciencias Médicas, Facultad de Medicina, Universidad Castilla-La Mancha (UCLM), 02008 Albacete, Spain; pjtarraga@sescam.jccm.es; 7Departamento de Expresión Plástica, Musical y Dinámica, Facultad de Educación, Universidad de Murcia (UM), 30100 Murcia, Spain; jlyuste@um.es

**Keywords:** children, feeding patterns, life style, physical fitness, sedentary behavior

## Abstract

Background: Studies have reported the association between cardiorespiratory fitness and higher adherence to the Mediterranean diet as well as lower recreational screen time. Similarly, higher screen time has been negatively linked to a lower adherence to the Mediterranean diet. However, the mediator effect of cardiorespiratory fitness on the influence of screen time on adherence to the Mediterranean diet is still unknown. The aim of this study was two-fold: first, to assess the combined association of recreational screen time and cardiorespiratory fitness with adherence to Mediterranean diet among Spanish schoolchildren, and second, to elucidate whether the association between recreational screen time and adherence to the Mediterranean diet is mediated by cardiorespiratory fitness. Methods: A descriptive and cross-sectional study was conducted. A total of 370 schoolchildren aged 6–13 years from six schools in the Region of Murcia (Spain) were included. Results: The mediation analysis showed that once screen time and cardiorespiratory fitness were included together in the model, cardiorespiratory fitness was positively linked to adherence to the Mediterranean diet (*p* = 0.020) and although screen time remained negatively related to adherence to the Mediterranean diet, this association was slightly attenuated (indirect effect = −0.027; 95% CI = (−0.080, −0.002)). Conclusions: This research supports that cardiorespiratory fitness may reduce the negative association between screen time and Mediterranean dietary patterns.

## 1. Introduction

Recreational screen time (ST) is considered as time spent watching screens (computer, mobile devices, and television) for reasons other than those associated with work or school [1]. The scientific literature indicates that higher time spent on recreational ST was linked to unfavorable health [2]. Thus, due to its relationship with adverse health outcomes, the World Health Organization (WHO) has recently suggested that youths should decrease the time engaged in sedentary behaviors, namely recreational ST [1].

In line with the above, Thivel et al. [3] showed in a study performed in 12 countries that children who spent a higher amount of ST presented an unhealthier dietary pattern. Similarly, children who had more ST ate fewer vegetables and fruits and tended to eat more snacks, energy drinks, and fast food, thus resulting in greater energy consumption [4]. Moreover, studies stated that children consume a large part of their daily caloric intake while engaged in recreational ST [5]. This finding could be at least partially explained by the amount of time engaged in recreational ST and the types of calorie-dense beverages and foods that are eaten while watching TV, as mass media operates as an incentive to eat, even distorting feelings of satiety or fullness [6]. Thus, the sum of these factors could lead to an overconsumption of food [7] and, consequently, to the development and maintenance of excess weight in youths [5,7].

As for healthy diet, the Mediterranean Diet (MD) has been recognized worldwide as a healthy dietary pattern due to its distinctive features [8]. It is a semi-vegetarian diet based on the daily intake of seasonal fruit and vegetables, olive oil for cooking and salad dressing, bread and whole milk, frequent intake of cheese and fish in variable amounts (depending on the distance to the sea), eggs, nuts, moderate wine consumption, and some meat (e.g., chicken, rabbit, lamb, and goat) [9]. Supporting this notion, scientific evidence has shown an inverse association of the MD with non-communicable diseases (e.g., hypertension, cancer, cardiovascular disease, or metabolic syndrome) [10], as well as mortality [11], emphasizing that some of its above-mentioned dietary components mainly influence these relationships [11]. In youths specifically, a greater adherence to the MD (understood as the following of some attitudes challenging and sustaining the conventional healthy Mediterranean dietary patterns) [8] was associated with higher anti-inflammatory potential properties [12]. Nevertheless, despite being an evidence-based healthy dietary pattern, a systematic review indicated the clear trend toward the abandonment of the MD in the countries from the Mediterranean region from 2004 to 2014, particularly amongst children [13].

Physical fitness is understood as muscular strength, balance, agility, and cardiorespiratory fitness (CRF), with CRF being associated with health outcomes [14]. Furthermore, a recent systematic review with metanalysis suggested that CRF level during childhood and adolescence, and its improvement, could be linked to a lower probability of suffering cardiometabolic disease and obesity in adulthood [15]. This is why childhood and adolescence periods may exert an essential function in the achievement of lifestyle habits, because being physically active, having an adequate level of physical fitness, and following a healthy diet are significant factors of present and future health [14].

Similarly, the compounded effect of several lifestyle-related behaviors could be superior to the summation of their separate effect sizes [16]. For this reason, a positive association has been found in several studies when analyzing the association between CRF and adherence to the MD [17,18]. Conversely, Carson’s systematic review showed that lower CRF was positively associated with higher ST [2]. Higher levels of ST were negatively linked to a higher adherence to the MD [7]. Thus, lifestyle factors are key to improving the likelihood of maintaining an active lifestyle as these factors cooperate with each other synergistically [19].

Accordingly, it was hypothesized that CRF could exert a mediator role in the association between ST and the MD. To prove this hypothesis, mediation analysis is a feasible statistical approach since it allows testing of the association between two variables, and to what degree a third variable may confound, mediate, or moderate that association [20]. In this way, researchers have analyzed the association between two variables through the mediator effect of CRF among schoolchildren, for instance, on the influence of high levels of body fat on both cardiometabolic risk and academic achievement [21], on the effect of diet on obesity [22], or even on the impact of excess of weight on cognition [23]. However, the mediator effect of CRF on the influence of ST on adherence to the MD is still unknown. Thus, this is the first study (to the best of our knowledge) to examine the mediator effect of CRF in the relation between ST and adherence to the MD.

Hence, the aim of this study was two-fold: first, to evaluate the combined association of recreational ST and CRF with adherence to the MD in Spanish schoolchildren, and second, to elucidate whether CRF is a mediator of the association between recreational ST and adherence to the MD.

## 2. Materials and Methods

### 2.1. Design and Participants

This cross-sectional study was carried out in the Valle de Ricote, which is composed of five different municipalities and is located in the Region of Murcia (Spain). All schools in these municipalities were asked to participate in this research. Participants were recruited by convenience sampling from six schools (public and private). Despite using this sampling method, all children from the eligible schools were requested to participate. The final sample included 370 Spanish schoolchildren aged 6–13 years. In order to participate in the study, participants were given a written informed consent that had to be signed by their parents or legal guardian. Additionally, these children and their parents or legal guardians received a short explanation about the study’s aims, and the tests and questionnaires that would be administered. In terms of inclusion criteria, only schoolchildren aged 6–13 years with parents or legal guardians who signed the informed consent were enrolled. Conversely, regarding the exclusion criteria, schoolchildren were not enrolled when they: (a) were exempt from the subject of physical education at school, since both the tests and the fulfilment of the questionnaires were carried out in the physical education lessons; (b) suffered from some kind of dysfunction that limited the practice of physical activity (i.e., any disease or motor problem); and (c) were taking some kind of pharmacological treatment. This study obtained ethics approval from the Bioethics Committee of the University of Murcia (ID 2218/2018). It was conducted according to the Helsinki Declaration and respected the human rights of the participants involved.

### 2.2. Procedures

Age and sex were self-reported. Area of residence was categorized into (1) rural (≤5000 inhabitants) and (2) urban (>5000 inhabitants) [24]. The type of schooling was dichotomized into two categories: (1) public and (2) private with public funds.

CRF was evaluated with the 20 m Shuttle Run Test, which has shown moderate-to-high validity for predicting maximum oxygen uptake and appears to be a helpful alternative for assessing this physical fitness component when it is not feasible to measure or determine the maximum oxygen uptake directly in a laboratory test [25]. In this test, children have to run between two lines 20 m apart while maintaining the rhythm with audio prompts from a Bluetooth-enabled speaker. The speed started at 8.5 km/h and this was augmented by 0.5 km/h per stage (with a stage duration of 1 min). The test ended when the schoolchildren did not reach the ending line at a time when the audio signals were received. CRF, in terms of maximal oxygen consumption (VO_2_max) (mL/kg/min), was predicted using a well-recognized protocol [26]. Additionally, we decided to perform the analyses by establishing two categories for CRF (low CRF = first and second tertile; high CRF = third tertile).

Recreational ST was measured based on the average daily hours that the children usually played videogames or watched TV. The children were asked for the number of hours that they usually engaged in recreational ST through the Krece Plus Short Test, which was previously validated by the enKid Study for participants aged 4–14 years [27]. The question used was: “On average, how many hours do you play video games or watch TV daily?”. As it is stated in this test, children from 6 to 12 years old answered by themselves with the help of the research assistant, while 13-year old participants answered alone. Likewise, two groups for recreational ST were created (≥2 h = high ST; <2 h = low ST) following the American Academy of Pediatrics guidelines for ST [28].

The KIDMED index was applied to test the adherence to the MD [8]. This questionnaire contains 16 questions associated with the attitudes of Mediterranean dietary patterns. The final score varies from −4 to 12 points, and it contains items with negative implications, which are scored with −1 point (e.g., skipping breakfast and consuming sweets or fast food). Items with positive implications are scored with +1 point (e.g., consumes fish regularly or has breakfast), as indicated previously.

Children were weighed twice, while wearing light clothing, using a digital scale (Tanita BC-545, Tokyo, Japan) and a stadiometer (Leicester Tanita HR 001, Tokyo, Japan). The *z*-scores for body mass index were calculated by the LMS method, according to the WHO criteria [29]. Waist circumference was measured with flexible metric tape at the intersection between iliac crest and the last rib. Skinfold measurements were taken at the triceps, biceps, iliac crest, and subscapular on the basis of the suggestions of the International Society for the Advancement of Kinanthropometry (ISAK). All measurements were performed by staff certified by the ISAK (level II). The full skinfold measurement computation was used to determine body density [30] and the Siri formula was considered to estimate body fat through body density [31].

### 2.3. Statistical Analysis

Continuous information is specified as the mean (standard deviation), while categorical information is indicated as numbers (percentages). The assumption of normality of the variables was verified by statistical procedures (Kolmogorov–Smirnov test), as well as graphical procedures (normal probability plot). We applied Student’s *t*-test to verify the differences in continuous information between sexes. CRF, recreational ST, and KIDMED score did not meet the assumption of normality and were converted before the analyses. To help with understanding, we used the two-step technique to transform non-normally distributed continuous data into normally distributed data [32]. Additionally, we created four different categories related to recreational ST and CRF to display the differences across them: (1) Low ST/High CRF, (2) High ST/High CRF, (3) Low ST/Low CRF, and (4) High ST/Low CRF. Bivariate and partial correlation coefficients (*r*), adjusted by potential covariates (age, sex, and body fat percentage), were calculated as a preliminary analysis to assess the relationships between CRF, recreational ST, and adherence to the MD. To assess the differences between the mean values of KIDMED score across categories of CRF (high and low) and recreational ST (high and low), analyses of covariance (ANCOVA) were applied. Pairwise post hoc comparisons were tested by the Bonferroni test. With a total sample of 370 schoolchildren, the posteriori sample calculation for ANCOVA showed a power of analysis of 98% probability of refusing the null hypothesis. Initial analyses revealed no significant interactions between sex and mean differences in recreational ST and CRF (ST *p* = 0.142; and CRF *p* = 0.994), as well as for age (ST *p* = 0.220; and CRF *p* = 0.466); hence, all analyses were carried out with both girls and boys together to increase statistical power.

Linear regression models were fitted using bootstrapped (10,000 samples) mediation techniques contained in the PROCESS SPSS macro (version 3.5), to inspect whether the association between recreational ST and adherence to the MD was mediated by CRF [33]. In this sense: (a) the regression of the mediator (CRF) on the independent variable (recreational ST) was applied in the first equation; (b) the regression of the independent variable (recreational ST) on the dependent variable (adherence to the MD) was performed in the second equation; (c) the regression of the adherence to the MD dependent variable on both the independent (recreational ST) and the mediator variable (CRF) was carried out in the third equation. The next criteria were considered to determine mediation: (i) the independent variable is linked to the mediator; (ii) the independent variable is linked to the dependent variable; (iii) the mediator is linked to the dependent variable; (iv) the association between the independent and dependent variable is diminished when the mediator is integrated in the regression model. Age, sex, area of residence, type of schooling, and body fat percentage were included as potential covariates. Statistical analyses were performed with SPSS 24.0 (IBM Corp, Armonk, New York, NY, USA), and statistical significance was considered at *p* < 0.050.

## 3. Results

Table 1 shows the descriptive data of the study participants. Girls reported both lower height and waist circumference than boys (*p* < 0.050). No statistically significant differences were found between boys and girls for KIDMED score, recreational ST, and CRF.

Table 2 reports the bivariate and partial correlations among recreational ST, adherence to the MD (KIDMED score), and CRF. CRF was positively linked to KIDMED score (*r* = 0.126) and inversely related to recreational ST (*r* = −0.088) (only when adjusted by potential confounders). Likewise, a negative relationship was found between recreational ST and KIDMED score (*r* = −0.171). Notwithstanding, all of these correlations were low.

ANCOVA showed the greatest adherence to the MD in a high CRF/low ST group (M = 7.2; SE = 0.1), whereas the lowest adherence to the MD was found in the low CRF/high ST group (M = 5.6; SE = 0.1). Likewise, we found a significant difference between high CRF/low ST and the other three groups: high CRF/high ST, low CRF/low ST, and low CRF/high ST (*p* < 0.050 for all) for KIDMED score after adjusting for age, sex, area of residence, type of schooling, and body fat percentage (Figure 1).

In general, when we verified the mediator role of CRF in the association between recreational ST and adherence to the MD, recreational ST was inversely related to CRF (*p* = 0.037) (first equation). Recreational ST was inversely related to adherence to the MD (*p* = 0.003) (second equation). Lastly, when recreational ST and CRF were integrated together in the regression model (third equation), CRF was positively related to adherence to the MD (*p* = 0.020) and although recreational ST continued to be inversely related to adherence to the MD, this relationship was slightly attenuated (indirect effect = −0.027; CI 95% = (−0.080, −0.002)). These results suggest that the association between recreational ST and adherence to the MD is mediated by CRF (Figure 2).

## 4. Discussion

Our study aimed to examine the combined relationship between recreational ST and CRF with adherence to the MD among Spanish schoolchildren as well as elucidate whether the relationship between recreational ST and adherence to the MD is mediated by CRF. The current results showed that children with higher CRF and lower recreational ST had the highest levels of adherence to the MD (KIDMED score). Likewise, the mediation analyses disclosed that the association between recreational ST and adherence to the MD was slightly attenuated by CRF. Therefore, a reduction in recreational ST should be accompanied by physical activity that generates improvements in CRF because children with a higher CRF (and less recreational ST) are more likely to adhere to the MD.

A negative association between CRF and recreational ST was found. This is consistent with the scientific literature, which indicated that increased engagement in recreational ST could reduce CRF levels, even when adjusting for physical activity [2,34]. Moreover, another systematic review showed that greater ST was significantly related to lower CRF across study designs, as well as some longitudinal studies [2]. Likewise, one systematic review with a metanalysis [35] found convincing evidence of the inverse association between recreational ST and CRF in children. One explanation for this finding could be related to displacement theory, which indicates that ST displaces the time spent in physical activities. This hypothesis has been proved in a longitudinal study, where it was found that children who exceed the amount of ST (greater than 2 h daily) at the age of six had higher body mass values and were less active at the age of eight and ten than their counterparts who at the age of six spent less ST [36].

In this study, a positive link between CRF and adherence to the MD was identified. These results agree with the results obtained by García-Hermoso et al. [37] in a recent systematic review and metanalysis with 565,421 youths, which indicated a positive relationship between MD and CRF. One reason that could justify this finding to certain extent is the above-mentioned association between MD and active lifestyle, because greater engagement in physical activity (as a consequence of active lifestyle) may increase the CRF [38]. It is well-known that part of the health benefits associated with the MD may be due to other properties of the Mediterranean lifestyle including physical activity. However, recently, it was indicated that children with higher CRF described a greater consumption of fats, proteins, and carbohydrates [22]. Likewise, earlier research showed higher CRF in children who presented moderate or great adherence to the MD [17] and in children with a greater frequency of vegetable and fruit consumption [39].

In the scientific literature, it is suggested that food consumption (especially if it takes the form of energy-dense snacks and beverages) could increase during TV viewing, which could increase total energy intake and, consequently, turn into body weight gains. This hypothesis is known as a mindless eating [40]. Likewise, recreational ST was related to a lower adherence to the MD in children [37]. Among the causes of the negative trend in adherence to the Mediterranean dietary pattern in Spanish youths, spending more than 4 h daily ST seemed to be the most relevant [41]. Similarly, our study also indicated an inverse relationship between recreational ST and adherence to the MD, concurring with the recent systematic review and metanalysis mentioned above [37]. Shi et al. [42] suggested that children who consume healthy foods are more likely to engage in physical activities and less likely to engage in sedentary behaviors in comparison with children who consume less healthy foods. Nonetheless, it is uncertain if children with more hours of ST have an overall unhealthy diet [43] or if they consume foods during ST that affect their general diet [44].

Finally, our findings suggest that the negative association of ST with the adherence to the MD was attenuated by CRF. This finding could be explained by two essential aspects. Firstly, though a large part of the variability in CRF is determined by genetics, certain lifestyle factors (especially physical activity) are also fundamental factors of CRF [45]. Along this line, higher CRF levels could diminish the negative trend of the above-mentioned displacement theory as far as recreational ST is concerned. Secondly, as a consequence of having an active lifestyle, and subsequently less time to spend in front of screens, children could have greater levels of CRF [46], which could mitigate the deleterious effects of the above-mentioned mindless eating hypothesis. Thirdly, another possible explanation could be some personality attributes, such as self-regulation. In this sense, children considered active (and a priori with higher CRF) and those who with low recreational ST were separately and jointly linked to higher self-regulation [47], which could be related to better eating habits. These results may provide essential information for healthcare professionals to promote effective interventions aimed at improving the healthy eating habits in children, contributing to the urgent need to improve healthy eating habits worldwide [48]. Furthermore, a healthy lifestyle during childhood should be promoted in a holistic manner (i.e., lower recreational ST and higher CRF through physical activity), rather than treated in isolation [49]. However, future experimental studies with randomization to manipulations in recreational ST exposure and CRF improvements are needed to prove cause–effect associations, test interventions, and establish mediators (mechanisms) of these associations.

In this study, some limitations should be reported. Firstly, because of the cross-sectional design of the current study, it was not possible to determine if the perceived associations indicate cause and effect relations. Secondly, we did not use a gas-analyzed peak oxygen uptake to assess CRF, though it is considered the physiological criterion for measuring children’s CRF. However, recommendations suggest the use of 20 m Shuttle Run Test to assess CRF as an international population health surveillance measure to offer further information on the health of pediatric populations [50]. Thirdly, another limitation is related to the lack of data on the individual’s timing of maturation (e.g., Tanner Stages), which could have provided a more accurate level of CRF [51]. Furthermore, self-report data were used to evaluated recreational ST in this study. Conversely, with regard to the strengths of our study, this is, to the best of our knowledge, the first study to analyze the mediator role of CRF in the relation between ST and adherence to the MD.

## 5. Conclusions

The present findings suggest that having a high CRF and less recreational ST are associated with higher adherence to the MD. Furthermore, the mediation analysis indicated that CRF mediates the relationship between recreational ST and adherence to the MD, slightly weakening this association. This knowledge could be essential for the appropriate and specific design of public health interventions that will contribute to the early adoption of Mediterranean dietary patterns. Likewise, these data indicate that intervention programs aimed at reducing recreational ST use and increasing CRF may be important to promote healthy dietary patterns.

## Figures and Tables

**Figure 1 ijerph-18-04490-f001:**
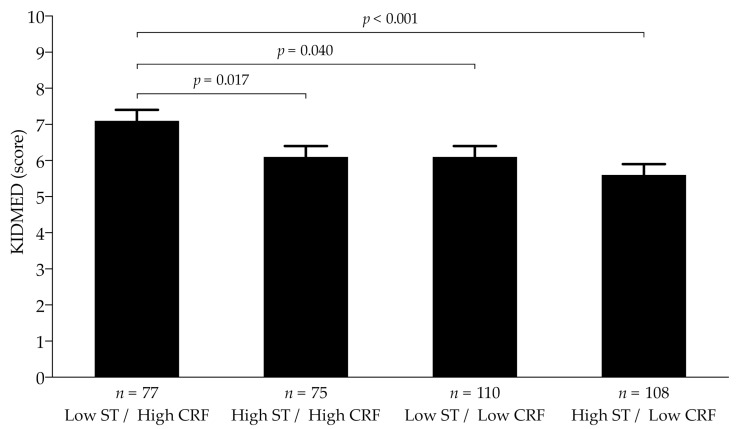
Combined effects of cardiorespiratory fitness and recreational screen time on adherence to the Mediterranean diet in schoolchildren. Estimated mean (dots) and 95% CIs (error bars) represent values after adjustment for age, sex, area of residence, type of schooling, and body fat percentage. CRF, cardiorespiratory fitness; ST, screen time.

**Figure 2 ijerph-18-04490-f002:**
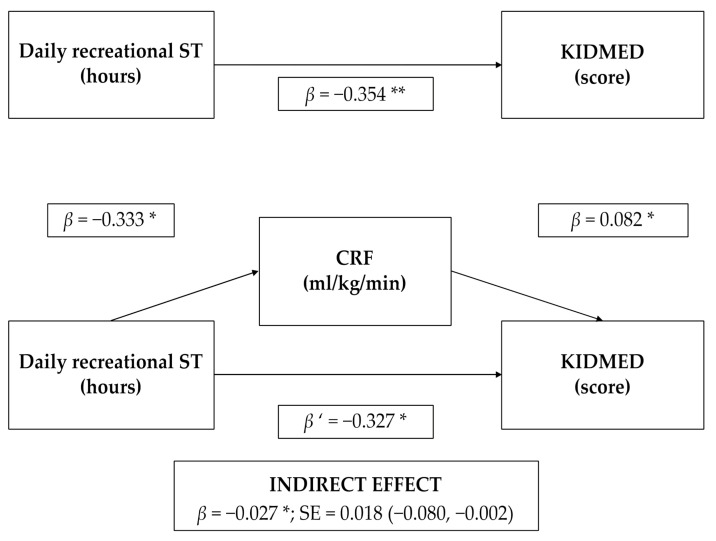
Cardiorespiratory fitness mediation model of the relationship between recreational screen time and adherence to the Mediterranean Diet in schoolchildren. CRF, cardiorespiratory fitness; ST, screen time. Adjusted by sex, age, type of schooling, area of residence, and body fat percentage. * *p* < 0.050; ** *p* < 0.001.

**Table 1 ijerph-18-04490-t001:** Descriptive characteristics of the analyzed sample stratified by sex.

Variables	Boys (*n* = 204)		Girls (*n* = 106)		*p*
	Mean/*n*	SD/%	Mean/*n*	SD/%	
Age (years)	8.8	1.8	8.5	1.8	0.084
Area of residence					
Rural	46	22.5	42	25.3	0.532
Urban	158	77.5	124	74.7	
Type of schooling					
Public	139	68.1	108	65.1	0.533
Private ^a^	65	31.9	58	34.9	
Anthropometric data					
Weight (kg)	36.41	10.75	34.9	11.10	0.178
Height (cm)	1.37	0.11	1.34	0.12	0.020
Body mass index (*z*-score)	1.14	1.25	1.15	1.20	0.909
Overweight/Obese ^b^	114	55.9	80	48.2	0.145
Waist circumference (cm)	63.1	7.7	60.8	8.63	0.009
Body fat percentage (%)	25.94	9.27	27.66	7.67	0.051
Physical fitness					
CRF (mL/kg/min)	45.34	4.39	44.62	3.92	0.101
Daily recreational ST					
Time spent (hours)	1.7	1.0	1.6	0.9	0.351
Adherence to the MD					
KIDMED (score)	6.1	2.1	6.3	2.0	0.476

Data are expressed as mean and SD or number and %. ^a^ Private with public funds. ^b^ Excess weight (overweight + obesity) computed according to the World Health Organization criteria [27]. CRF, cardiorespiratory fitness; KIDMED, Mediterranean diet quality index for children and teenagers; MD, Mediterranean diet; ST, screen time.

**Table 2 ijerph-18-04490-t002:** Bivariate and partial correlations among cardiorespiratory fitness, recreational screen time, and adherence to the Mediterranean Diet.

Variables	CRF (mL/kg/min)		Daily Recreational ST (hours)		KIDMED (Score)	
	Crude	Adjusted ^#^	Crude	Adjusted ^#^	Crude	Adjusted ^#^
CRF (mL/kg/min)	-	-	−0.050	−0.088 *	0.140 *	0.126 *
Daily recreational ST (hours)	-	-	-	-	−0.163 *	−0.171 **

CRF, cardiorespiratory fitness; KIDMED, Mediterranean diet quality index for children and teenagers; ST, screen time. ^#^ Adjusted by sex, age, area of residence, type of schooling, and body fat percentage. * *p* < 0.050; ** *p* < 0.001.

## Data Availability

The data presented in this study are available on request from the corresponding author. The data are not publicly available because they belong to minors.

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
