# Peer review of "Cardiorespiratory Fitness as Mediator of the Relationship of Recreational Screen Time on Mediterranean Diet Score in Schoolchildren"

_ijerph, 2021, doi:10.3390/ijerph18094490_

Round 1
Reviewer 1 Report
This is an interesting descriptive and cross-sectional study that aimed to assess firstly the association between screen time and adherence to Mediterranean diet among Spanish school children population, and secondly, to elucidate whether this association is mediated by cardiorespiratory fitness. English language should be carefully revised by a native speaker; I also suggest to check throughout the text for spelling errors and consistent use of abbreviations. The Methods section should be improved. Information on length of follow up (if any), ethical committee approval, and inclusion and exclusion criteria should be added. I would also suggest to include more information and description of Mediterranean Diet (e.g. specific foods and quantity, etc). The figures and tables are adequate. I would also suggest to include further discussion on the future direction and possible clinical application of the results.
Author Response
This is an interesting descriptive and cross-sectional study that aimed to assess firstly the association between screen time and adherence to Mediterranean diet among Spanish school children population, and secondly, to elucidate whether this association is mediated by cardiorespiratory fitness.
Thank you very much for your comment.
English language should be carefully revised by a native speaker; I also suggest to check throughout the text for spelling errors and consistent use of abbreviations.
We have applied a consistent use of the abbreviations. Similarly, the manuscript has been revised by a native English speaker.
The Methods section should be improved. Information on length of follow up (if any), ethical committee approval, and inclusion and exclusion criteria should be added.
Thanks for your indication. This is a cross-sectional study (without follow up). Information about ethical committee approval has been previously indicated in Lines 127-130: “This study obtained ethics approval by the Bioethics Committee of the University of Murcia (ID 2218/2018)”. Similarly, information about exclusion criteria appeared previously, however, we have modified them: “Conversely, regarding the exclusion criteria, schoolchildren were not enrolled when: a) were exempt from the subject of Physical Education at school, since both the tests and the fulfilment of the questionnaires were carried out in the Physical Education lessons; b) suffered from some kind of dysfunction that limited the practice of physical activity (i.e., any disease or motor problem); and c) were taking some kind of pharmacological treatment”. Similarly, we have added information about inclusion criteria (Lines 182-184): “In terms of inclusion criteria, only schoolchildren aged 6-13 years, whose parents/legal guardians signed the informed consent, were enrolled.”
On the other hand, information about design and participants has been re-structured to provide a better understanding.
I would also suggest to include more information and description of Mediterranean Diet (e.g. specific foods and quantity, etc).
Thank you. We have added the next information (Lines 61-65): “In this sense, this is a semi-vegetarian diet, based on the daily intake of seasonal fruit and vegetables, olive oil for cooking and salad dressing, bread and whole milk, frequent intake of cheese, fish in variable amounts (depending on the distance to the sea), eggs, nuts, moderate wine consumption, and some meat (e.g., chicken, rabbit, lamb, goat) [1]”.
The figures and tables are adequate.
Thank you.
I would also suggest to include further discussion on the future direction and possible clinical application of the results.
According to your indication, we have added the following information (Lines 339-346): “The present findings suggest that having a high CRF and spending less time in recreational ST is associated with higher adherence to DM. Furthermore, the mediation analysis indicates that CRF mediates the relationship between recreational ST and adherence to the MD, slightly weakening this association. This knowledge could be essential for the appropriate, specific design of public health interventions that will contribute to early adoption of Mediterranean dietary patterns”.
References
- Radd-Vagenas, S.; Kouris-Blazos, A.; Singh, M.F.; Flood, V.M. Evolution of Mediterranean Diets and Cuisine: Concepts and Definitions. Asia Pacific Journal of Clinical Nutrition 2017, 26, doi:10.6133/apjcn.082016.06.

Reviewer 2 Report
21-22, what?
What is ST and MD¿?
Delete M & SD from the abstract.
Abstract needs to be fully rewritten… I really do not understand most of it, and I doubt that readers do so.
112-113 ¿?
two categories for CRF (low 115 CRF = first and second tercile 1; high CRF = third tercile). What is the criteria for this¿?
From this point I stopped reading, and made a diagonal review of the article.
At the level of scientific writing it lacks format and form, at the level of writing and English grammar, the level is very low, it needs to be seriously reviewed by a native external.
A lot of typographical errors which denote speed when writing the article.
At the methodological level, a vast amount of details are lacking, such as the inclusion and exclusion criteria, elements referring to the procedures in data collection.
The statistical treatment is doubtful the correct one. Furthermore, there should be a specific point in the text about this section and not integrated.
Another interesting element is the diversity of authors who participate in the study, being the area and region of study local (Murcia and conquretly a village). For a study of such simplicity and local, is it necessary a total of 5 authors? And how is the membership of these authors justified in countries like Chile or Urugay?
At a general level, the relationships that are presented are causal, they do not have scientific weight and the sample size is insufficient to make these conclusions. Moreover, the socioeconomic and sociodemographic profile is a more than influential factor in the variables which the authors address, a fact that I am still aware of if they have read García-Hermoso. This section has to be included in the sample section, and is strictly necessary. Or does the rural population have a behavior similar to that of the city?
Finally, the article lacks sufficient quality and significance to be published in a journal of this profile.
I recommend the authors prudence with the authorship
Author Response
21-22, what?
It has been modified, thanks: “Studies have pointed out the association between cardiorespiratory fitness and higher adherence to the Mediterranean diet, as well as lower screen time. Similarly, spending higher screen time have been negatively linked to a lower adherence to the Mediterranean diet”.
What is ST and MD¿?
Screen time and Mediterranean Diet. We have added this information.
Delete M & SD from the abstract.
It has been deleted, thank you.
Abstract needs to be fully rewritten… I really do not understand most of it, and I doubt that readers do so.
Thank you. It has been fully modified.
112-113 ¿?
It has been modified as follows: “Hence, the aim of this study was two-fold: first, to evaluate the combined association of recreational ST and CRF with adherence to MD in Spanish schoolchildren, and second, to elucidate whether CRF is a mediator of the association between recreational ST and adherence to the MD”.
Two categories for CRF (low 115 CRF = first and second tercile 1; high CRF = third tercile). What is the criteria for this¿?
In absence of well-established cut-off points of CRF, we decide to calculate the different tertiles to obtain a balanced split of the sample. We would like to emphasize how higher levels of CRF could be related with both MD and ST. Moreover, this choice was made to increase the statistical power.
From this point I stopped reading, and made a diagonal review of the article.
Thank you.
At the level of scientific writing it lacks format and form, at the level of writing and English grammar, the level is very low, it needs to be seriously reviewed by a native external. A lot of typographical errors which denote speed when writing the article.
Thank you. The manuscript has been revised by a native English speaker and some typographical errors have been changed.
At the methodological level, a vast amount of details are lacking, such as the inclusion and exclusion criteria, elements referring to the procedures in data collection.
OK, thank you. The exclusion criteria have been fully explained (Lines 123-128): “Conversely, regarding the exclusion criteria, schoolchildren were not enrolled when: a) were exempt from the subject of Physical Education at school, since both the tests and the fulfilment of the questionnaires were carried out in the Physical Education lessons; b) suffered from some kind of dysfunction that limited the practice of physical activity (i.e., any disease or motor problem); and c) were taking some kind of pharmacological treatment”. Similarly, we have added information about inclusion criteria (Lines 121-122): “In terms of inclusion criteria, only schoolchildren aged 6-13 years, whose parents/legal guardians signed the informed consent, were enrolled”.
On the other hand, information about procedures in data collection already appeared in the manuscript (Lines 111-117): “This cross-sectional study was carried out in the Valle de Ricote, which is composed of five different municipalities and is located in the Region of Murcia (Spain). All the schools of these municipalities were requested to take part in this research. Participants were recruited by a convenience sample from six schools (publics and privates). In spite of using this sampling method, all children from the eligible schools were requested to participate. The final sample included 370 Spanish schoolchildren aged 6–13 years”.
Additionally, information about design and participants has been re-structured to provide a better understanding.
The statistical treatment is doubtful the correct one. Furthermore, there should be a specific point in the text about this section and not integrated.
This statistical procedure has been used in several publications [1–3]. All of these procedures could be found in Hayes [4]. Similarly, a specific detailed explanation about this statistical can be found in the manuscript (Lines 196-208).
Another interesting element is the diversity of authors who participate in the study, being the area and region of study local (Murcia and conquretly a village). For a study of such simplicity and local, is it necessary a total of 5 authors? And how is the membership of these authors justified in countries like Chile or Urugay?
Thanks for your reflection. As the reviewer could check out, we are a group of researchers located in different parts of the world who are working together. All the co-authors meet the scientific criteria to be added as co-authors to this manuscript.
Besides, a couple of co-authors have worked in Latin-America and they have still double affiliation. Their contributions to the current paper are part of their work with these institutions and this is the reason why they have decided to add the second affiliation.
At a general level, the relationships that are presented are causal, they do not have scientific weight and the sample size is insufficient to make these conclusions.
Thank you for your comment. Since the design of the study is cross-sectional, we cannot establish cause-effect associations. This is a limitation and it has been already added to the discussion section. As suggested, we have softened our conclusions.
Finally, the article lacks sufficient quality and significance to be published in a journal of this profile. I recommend the authors prudence with the authorship
Thank you for your reflection. We do agree that study has some limitations (we have mentioned specifically in the discussion section), however, as far as we are concerned, this is the first study that analyzed the mediator effect of cardiorespiratory fitness in the relation between screen time and adherence to the Mediterranean diet. After the (possible) publication of this study, future studies could be performed to elucidate this association (as we indicate in the conclusions section).
References
- García-Hermoso, A.; Martínez-Vizcaíno, V.; Sánchez-López, M.; Recio-Rodriguez, J.I.; Gómez-Marcos, M.A.; García-Ortiz, L. Moderate-to-Vigorous Physical Activity as a Mediator between Sedentary Behavior and Cardiometabolic Risk in Spanish Healthy Adults: A Mediation Analysis. Int J Behav Nutr Phys Act 2015, 12, 78, doi:10.1186/s12966-015-0244-y.
- García-Hermoso, A.; Esteban-Cornejo, I.; Olloquequi, J.; Ramírez-Vélez, R. Cardiorespiratory Fitness and Muscular Strength as Mediators of the Influence of Fatness on Academic Achievement. J Pediatr 2017, 187, 127-133.e3, doi:10.1016/j.jpeds.2017.04.037.
- García-Hermoso, A.; Carrillo, H.A.; González-Ruíz, K.; Vivas, A.; Triana-Reina, H.R.; Martínez-Torres, J.; Prieto-Benavidez, D.H.; Correa-Bautista, J.E.; Ramos-Sepúlveda, J.A.; Villa-González, E.; et al. Fatness Mediates the Influence of Muscular Fitness on Metabolic Syndrome in Colombian Collegiate Students. PLoS One 2017, 12, e0173932, doi:10.1371/journal.pone.0173932.
- Hayes, A.F. Introduction to Mediation, Moderation, and Conditional Process Analysis: A Regression-Based Approach; Methodology in the social sciences; Second edition.; Guilford Press: New York, 2018; ISBN 978-1-4625-3465-4.

Reviewer 3 Report
Congrats. The work is good, and has been correctly conducted by the authors, which is positively valued. With the intention of trying to improve it, a couple of considerations are proposed to take into account.
Lines 21-24: its is necessary to rewrite.
Lines 106-109: The concepts "assessment", "estimation" and "directly measured" should be used with greater precision. I suppose that what the authors have done is to "estimate" the maximum oxygen consumption, with this test, since it appears to be a helpful alternative when it is not feasible to "measure or determine directly" in a laboratory. I recommend specifying it.
Author Response
Congrats. The work is good, and has been correctly conducted by the authors, which is positively valued. With the intention of trying to improve it, a couple of considerations are proposed to take into account.
Thank you so much for your feedback.
Lines 21-24: its is necessary to rewrite.
Thank you. It has been modified as follows (Lines 22-30): “Studies have pointed out the association between CRF and higher adherence to the MD, as well as lower ST. Similarly, spending higher ST have been negatively linked to a lower adherence to the MD. However, the mediator effect of the cardiorespiratory fitness on the influence of the recreational screen time on adherence to the Mediterranean diet is still unknown”.
Lines 106-109: The concepts "assessment", "estimation" and "directly measured" should be used with greater precision. I suppose that what the authors have done is to "estimate" the maximum oxygen consumption, with this test, since it appears to be a helpful alternative when it is not feasible to "measure or determine directly" in a laboratory. I recommend specifying it.
Ok, thanks. As suggested, we have used “predict” following the original reference published by Léger et al [1].
References
- Léger, L.A.; Mercier, D.; Gadoury, C.; Lambert, J. The Multistage 20 Metre Shuttle Run Test for Aerobic Fitness. Journal of Sports Sciences 1988, 6, 93–101, doi:10.1080/02640418808729800.

Round 2
Reviewer 1 Report
The Authors replied satisfactorily to all my comments
Academic Editor Notes
The editors consider that the article is accepted. But it should be included in the limitation section that the use of the course navette with young children has many limits and biases.
Reply to Academic Editor
Thank you for your feedback. You are right. Thus, we have included that limitation as follows:
"Secondly, we did not use a gas-analyzed peak oxygen uptake to assess CRF, since it is considered the physiological criterion for measuring children’s CRF. However, it has been recommended to consider the use of 20-m Shuttle Run Test to assess CRF as an international population health surveillance measure to offer further information on the health of the pediatric population [1]"
References
[1]. Tomkinson, G.R.; Lang, J.J.; Blanchard, J.; Léger, L.A.; Tremblay, M.S. The 20-m Shuttle Run: Assessment and Interpretation of Data in Relation to Youth Aerobic Fitness and Health. Pediatric Exercise Science 2019, 31, 152–163, doi:10.1123/pes.2018-0179.
Reviewer 2 Report
Authors have made substantial changes, still i feel that the quality of the research work is not enough for the present journal.
Academic Editor Notes
The editors consider that the article is accepted. But it should be included in the limitation section that the use of the course navette with young children has many limits and biases.
Reply to Academic Editor
Thank you for your feedback. You are right. Thus, we have included that limitation as follows:
"Secondly, we did not use a gas-analyzed peak oxygen uptake to assess CRF, since it is considered the physiological criterion for measuring children’s CRF. However, it has been recommended to consider the use of 20-m Shuttle Run Test to assess CRF as an international population health surveillance measure to offer further information on the health of the pediatric population [1]"
References
[1]. Tomkinson, G.R.; Lang, J.J.; Blanchard, J.; Léger, L.A.; Tremblay, M.S. The 20-m Shuttle Run: Assessment and Interpretation of Data in Relation to Youth Aerobic Fitness and Health. Pediatric Exercise Science 2019, 31, 152–163, doi:10.1123/pes.2018-0179.